

**Evaluation of microwave radiances of GPM/GMI for the all-sky**
**assimilation in RTTOV framework**
Rohit Mangla[1], Jayaluxmi Indu[1,2]
[1]Department of Civil Engineering, Indian Institute of Technology, Bombay, India
[2]Interdisciplinary Center for Climate Studies, Indian Institute of Technology Bombay, India
*Correspondence to*: Jayaluxmi Indu (indusj@civil.iitb.ac.in)
**Abstract.** This study evaluates the all-sky GPM/GMI radiances towards assimilation in regional
mesoscale model at $183\pm7$ GHz. The radiative transfer model (RTM) namely RTTOV-SCATT is
used for the simulation of three tropical cyclones (hudhud, vardah and kyant respectively). Within
the RTM, the performance of non-spherical Discrete Dipole Approximation (DDA) shapes (sector
snowflake, 6-bullet rosette, block-column and thinplate) are evaluated. The input data used in
RTTOV-SCATT includes vertical hydrometeor profiles, humidity and surface fluxes. In addition,
the first guess simulations from Weather Research Forecast (WRF) model were executed at 15 km
resolution using ERA-Interim reanalysis datasets. Results indicate that observed minus first guess
(FG departures) are symmetric with DDA shapes. The normalized probability density function of
FG departures shows large number of spatially correlated samples between clear-sky and poorly
forecasted region. Quality control (QC) method was performed to eliminate large FG departures
due to instrumental anomalies or poor forecast of clouds and precipitation. The goodness of fit
test, h-statistics and skewness of observed and simulated distribution show optimum results for
thinplate shape in all the convective events. We also tested the high resolution ERA-5 reanalysis
datasets for the simulation of all-sky radiances using thinplate shape. Results illustrate a potential
to integrate the GMI sensor data within a WRF data assimilation system.





## 1 Introduction


The Numerical weather prediction (NWP) model is widely used for forecasting the evolution of
the surface and atmospheric conditions. To predict the state of the atmosphere, the NWP model
relies on mathematical models and best initial conditions of the state of the atmosphere. Even
though NWP models provide meaningful forecasts, they are biased owing to model structure and
approximation of subgrid-scale processes (Shastri et al., 2017). In addition, it is also challenging
to define the best initial conditions of the atmosphere state. Recent developments by operational
NWP centers state that assimilating satellite radiances improves the forecast skills (Islam et al.,
2016; Routray et al., 2016; Saunders et al., 2013; Singh et al., 2016). Studies show that assimilation
of all-sky (clear and cloudy) microwave radiances in global NWP models have a large positive
impact on temperature and humidity (Geer, 2013; Kazumori et al., 2014; Lean et al., 2017). The
satellite radiances from microwave imagers [Tropical Rainfall Measuring Mission (TRMM)
microwave imager (TMI) (Kummerow et al., 1998), Aqua Advanced microwave scanning
radiometer for earth observing system (Kawanishi et al., 2003), Advanced Microwave Scanning
Radiometer-2 (AMSR-2) (JAXA, 2013), Special Sensor Microwave Imager Sounder (SSMIS)
(Kunkee et al., 2008) and Global Precipitation Measurement (GPM) Microwave Imager (GMI)
(Hou et al., 2014)] contain crucial information on deep and intense convection (Otkin, 2012).
Study of rainfall/convective systems involve examining the naturally emitted electromagnetic
radiation from the earth which interacts with atmospheric gases like water vapour, hydrometeors
(precipitation-sized particles of rainfall, snowfall, ice crystals etc). A radiative transfer model
(RTM) uses profiles of all the observed variables to provide the satellite observations which can
be either brightness temperature (Tb) (from radiometer) or reflectivity (from radar). An accurate
comparison between model and observed forecast relies strongly on the assumptions used for
radiation-hydrometeor interaction. To generate improved initial conditions of the model, a crucial
role is played by the assumptions made when simulating observations from different instruments
in space. To date, the scientific community has not really examined this important aspect.
Existing studies state that within the microwave frequency ranges (10-183 GHz), assuming
spherical shapes for snow/ice particles in RTM models produce un-realistic scattering in deep
convective clouds (Hong et al., 2005). Geer and Baordo, (2014) introduced the realistic 3D discrete
dipole approximation (DDA) non-spherical shapes to represent frozen hydrometeors within an



RTM model. The DDA shapes examined are usually long column, short column, block column,
thick plate, thin plate, 3,4,5 and 6 bullet rosette, sector and dendrite snowflake shapes (Liu, 2008).
Geer and Baordo, (2014) claimed that DDA sector snowflake is approximately fit for all
frequencies at a global scale but it doesn't perform well at the regional level. Regional case studies
show that block column over Indian ocean (Guerbette et al., 2016) and thinplate over
Mediterranean region (Rysman et al., 2016) have improved the simulation of low Tb at convective
scale for 183 GHz. Hence, careful investigation of DDA shapes is required at the regional level,
prior the simulation of cloudy radiances at a higher frequency.  Generally, the best choice of DDA
shapes at each frequency is based on the statistical analysis of the FG departures which is important
in variational data assimilation techniques (Fowler and Van Leeuwen, 2013). Assimilation of
satellite radiances offer difficulty due to cloud processes that non-linearly affect the upwelling
radiations, non-gaussian FG departures statistics and systematic biases from NWP and radiative
transfer models (RTM) (Errico et al., 2007; Okamoto, 2017).
The issue of non-gaussian characteristics of FG departure could be resolved using cloud dependent
standard deviation ($SD_{cloud}$) (Geer and Bauer, 2011; Okamoto, 2017; Okamoto et al., 2014). The
study conducted by Geer and Bauer, (2011) proposed a symmetric error model between $SD_{cloud}$
and cloud amount predictor for all-sky microwave observations. The authors used the error model
for AMSR-E observations at 19 GHz and observed that the probability distribution function (pdf)
of normalized FG departures follow a gaussian distribution if normalization is done by $SD_{cloud}$.
Geer, (2013) used the same model at multiple frequencies of TMI and SSMIS channel. In
microwave spectrum, the symmetric error model is known to perform well for low frequencies
(<50 GHz) as low frequencies are sensitive to cloud liquid droplets and rain-drops (Skofronick-
Jackson and Wang, 2000) for which the particle shape and density are pre-defined. At higher
microwave frequencies, the backscatter/brightness temperature registered by the sensor is mainly
due to scattering from frozen hydrometeors, assuming a spherical shape (Geer and Baordo, 2014).
Long-term monitoring of FG departure was found useful for identifying the instrumental error
from ground based microwave observation (De Angelis et al., 2017).
This study investigates the simulation of all-sky GMI radiances of tropical cyclones over Indian
region at $183 \pm 7$ GHz. In the analysis, we examined the cloud effect to evaluate the normalized
FG departures. For the appropriate selection of DDA shapes, we inspect the statistical measure of



FG departures. In addition, we also include the analysis of ERA-5 reanalysis datasets (Malardel et
al., 2015) to extend the sensitivity to cloud physical processes at a higher resolution. Section 2
briefly summarizes the GMI radiance datasets, NWP and RTM experimental setup. The simulation
results and error analysis are demonstrated in section 3. Summary and conclusions are provided in
section 4.

**2   Data and Methods**
**2.1 GPM/GMI observations**
The GMI sensor is a conically scanning passive radiometer on board the GPM satellite (Hou et al.,
2014) developed by National Aeronautics and Space Administration (NASA) in collaboration with
Japan Aerospace Exploration Agency (JAXA) and successfully launched on 28[th] February 2014.
Being a successor of TRMM (Kummerow et al., 1998) the GPM mission has several advantages.
GMI data is acquired in 13 channels in low (10-89 GHz) and high frequency (166-183 GHz) bands
[Table 1] while TMI was limited to just 9 low-frequency bands. GMI has additional capabilities
of detecting light precipitation and extending the global coverage to the mid-latitude region
$(60^o S - 60^o N)$. The horizontal resolution has been improved in GMI datasets because of the
increase in reflector size of GMI (1.2 m) from TMI (61 cm). Figure 1 shows hudhud cyclone event
on 9[th] October 2014, 06 UTC at 10, 89 and 183±7 GHz frequency. The low frequency channel (10
GHz) is sensitive to only liquid precipitation and greatly affected by surface emissivity (Hou et
al., 2014). The discrimination between land and ocean (Figure 1) is clear in 10 GHz, moderate in
89 GHz and insensitive at 183 GHz. Furthermore, $183 \pm 7$ GHz channel can investigate deeply
the atmosphere and it is highly sensitive to frozen hydrometeors. This channel is moderately
sensitive to rain and cloud liquid water and also detect the scattering signals from small ice
particles (Bennartz and Bauer, 2003; Laviola and Levizzani, 2011). In Figure 1, a strong
depression of temperature lower than 100 K can be seen over ocean at $183 \pm 7$ GHz indicating
the presence of frozen hydrometeors in deep convection. For the present study, GMI level 1b
radiances for three tropical cyclones at $183 \pm 7 \, GHz$-V (hereafter band 13) is used.






## 2.2 RTTOV-SCATT v12.1 Model

The all-sky GMI radiances have been simulated using RTTOV-SCATT (version 12.1) of Radiative Transfer for the television infrared observation vertical sounder (RTToV) (Hocking et al., 2017; Saunders et al., 2017). The RTToV is initially developed by ECMWF which was then upgraded within the European Organization for the Exploitation of Metereological Satellites (EUMETSAT) NWP satellite application facility. This Model is suitable for rapid transformation of a huge number of NWP model outputs into the radiance space. The RTTOV-SCATT is a separate interface for the simulation of cloud and precipitation affected microwave radiances. As an input to RTTOV-SCATT model, the atmopheric profiles (i.e. temperature, water vapour, cloud liquid water, ice, snow and rain) were derived from WRF NWP model output. The delta-eddington approximation is used for solving the radiative transfer equations to simulate the scattering effects of clouds and precipitations (Joseph et al., 1976). The surface emissivity over oceans are caluicated by the surface parameters (eg. Temperature, surface wind) using the microwave surface emissivity model (FASTEM-version 6) (Kazumori and English, 2015). The all-sky Tb computed represent the weighted summation of the clear and cloudy independent columns (eq. 1). The weighing criteria is decided by the cloud fraction (Geer et al., 2009) which is based upon the variation in cloud and precipitation at subgrid scale.

$$Tb^{all-sky} = C_f * Tb^{cloudy} + (1 - C_f) * Tb^{clear-sky} \qquad (1)$$

Here, $C_f$ represents the vertical profile of cloud fraction.

## 2.3 WRF NWP Model

The WRF is specifically designed for regional forecast in operational and research NWP centers (Skamarock et al., 2008). The present study used the version 3.8 of WRF model for the forecasting of tropical cyclones over Indian region. We designed the experimental setup in a single domain from $3^o N$ to $26^o N$ and from $73^o E$ to $103^o E$ with 213x165 horizontal grids of 15 km resolution [Figure 2 (a)]. This experiment is configured with 51 number of vertical layers and model top is at 125 hPa. The initial and boundary conditions are taken from ERA-Interim reanalysis datasets (product of ECMWF) with specification of 71 km spatial resolution at 6 h interval (Simmons et al., 2007). Geographical parameters including land use land cover (LULC), topography, soil type,



lake and vegetation parameters are provided by the United States Geological Survey (USGS)
global datasets at 30 sec resolution. Three tropical cyclones named "Hudhud" (October7-14,
2014), "Vardah" (December 6-12, 2016) and "Kyant" (October 21-27, 2016) over Bay of Bengal
(BOB) regions are considered in this study. Their tracks are shown in Figure 2 (b).
The physical parameterization schemes used are as suggested by (Routray et al., 2016) over BOB
region are; WRF single moment 6-class microphysics scheme (Hong and Lim, 2006), Kain-Fritsch
convection scheme (Kain, 2004), Yonsei scheme for planetary boundary layer (Hong et al., 2006),
Dudhia shortwave radiation scheme (Dudhia, 1989), rapid radiative transfer model scheme for
long-wave radiation (Mlawer et al., 1997) and Noah land surface model scheme (Tewari et al.,
2004). This configuration is highly versatile for the prediction of short range forecast over the
Indian region (Kumar et al., 2014; Singh et al., 2016).
**3    Results and Discussion**
In the present study, DDA shapes of sector snowflake is used as first step for intial error analysis
in section 3.1-3.4 (Geer and Baordo, 2014). In section 3.5, a statistical investigation is conducted
to identify the best shape among the recognized DDA shapes (i.e., Sector snowflake, thinplate, 6-
bullet rosette and block-column). In this study, the density of hydrometeros (rain and cloud liquid
water=1000 kg/m$^3$; ice=917 kg/m$^3$ ; snow= 50 kg/m$^3$) and particle size distributions by Field et al.,
(2007) for snow, marshall-palmer distribution for rain, modified-gamma distribution for cloud
liquid water and cloud ice have been used.
**3.1 Spatial Distribution of observed and simulated Tb**
The Figure 3 shows comparison between the all-sky simulated radiances at band 13 with respect
to the observed GMI radiances for three cyclonic events over the BOB region. The microwave
observations were averaged to 15 km horizontal resolution to match closely with the effective
resolution of NWP model. The increased scattering from frozen hydrometeors at band 13 in deep
convective zones results in low temperatures of observed radiances inside the core of cyclone (upto
70-80 K). Underestimation was observed using the mie-sphere, sector snowflake and six-bullet
rosette shapes. Though the overall pattern and location of convective clouds near the eye of cyclone
matched closely with the observations, Tb inside the core can be found to vary with hydrometeor
shapes and estimates. This may be attributed to deficiency of frozen hydrometeors at sub-grid scale





in Kain-Fritsch convection scheme (Rysman et al., 2016). A study by Wu et al., (2015) found that
frozen hydrometeors are underestimated in WRF simulations by all convective parametrization
schemes over central and eastern pacific region. Rysman et al., (2016) estimate the
underestimation in WRF simulations by a factor of 5 using airborne radar in Mediterranean region.
The Figure 4 shows distribution of FG departures in mie-spheres and DDA shapes for all the case
study events. A negative departure occurs when the RTToV model is unable to produce realistic
representations owing to cloud and precipitation. Within DDA shapes, the pdf curve is found to
follow a symmetric distribution. The error is spread equally in both directions due to random
forecast errors from first-guess and observations. In case of mie-spheres, the shift towards large
negative departures indicates the presence of bias in cloudy region. This is because of insufficient
scattering by mie-spheres at band 13 (Geer, 2013). Results show that DDA simulations provide a
better realistic scattering in all-sky conditions when FG departures are symmetrical in nature.

**188    3.2 Determination of observation errros with cloud amount**

The standard deviations of FG departures in clear-sky assimilation are referred to as observation
errors. The observation errors in all-sky radiance assimilation for microwave observations are
generally computed from symmetric error models (Geer and Bauer, 2011). Error models are a
function of cloud amount predictor at 37 GHz. In the present study, the observed/simulated cloud
amounts have been computed from observed/simulated radiances in clear and all-sky conditions.
$C_{37} = 1 - PD_{37};$            (2)
$PD_{37} = \frac{Tb_v - Tb_h}{Tb_v^{clr} - Tb_h^{clr}} \approx \tau_{cloudy}^2$            (3)
Where, $Tb_v$ & $Tb_h$ are the vertically and horizontal polarised radiances in cloudy condition; $Tb_v^{clr}$
and $Tb_h^{clr}$ are the vertically and horizontal polarized radiances in clear sky condition. $PD_{37}$ is the
normalized polarization differences approximately equal to square of transmittance in cloudy
region (Petty, 1994). A clear and cloud sky is represented using a $PD_{37}$ of 1 and 0 respectively.
For easy interpretation, we preferred cloud amount ($C_{37}$) which varies from 0 to 1 for the same
representation. As the quantities of $C_{37obs}$ and $C_{37sim}$ are affected with sampling error (Geer and
Bauer, 2011), their average is considered as the average cloud amount $C_{37avg}$.





Figure 5 shows the $SD_{cloud}$ curve at band 13 using the $C_{37avg}$ on x-axis in a bin range of 0.05.
At $C_{37avg} \sim 0$, both observations and first-guess are free from clouds (i.e. clear-sky condition). As
the $C_{37avg}$ increases, the error is found to initially increase linearly and attain the maxima at
$C_{37avg} \sim 0.48$ in all the meteorological events after which the error starts declining to the
maximum cloud amount ($C_{37avg} = 1$). The sudden peak at $C_{37avg} \sim 0.8$ observed in hudhud and
vardah cyclones was due to poor representation of Tb at higher frequency using DDA sector
snowflake shape in heavy clouds that causes large error.
In symmetric error model, the $SD_{cloud}$ curve was piecewise linearly transformed as a function of
$C_{37avg}$ (Geer and Bauer, 2011) (eq. 4).

$$SD_{cloud}\left(C_{37avg}\right) = \begin{array}{l} S_{clr} \; if \; C_{37avg} \leq C_{clr} \\ s_{clr} + \left(\frac{S_{cld}-S_{clr}}{C_{cld}-C_{clr}}\right)(C_{37avg} - C_{clr}) \; if \; C_{clr} < C_{37avg} < C_{cld} \\ S_{cld} \; if \; C_{37avg} \geq C_{cld} \end{array} \qquad (4)$$


Here, $S_{clr}$ is the minimum $SD_{cloud}$ defined by the threshold $C_{clr}$ in clear-sky region, whereas $S_{cld}$
is maximum $SD_{cloud}$ in strongly dominating clouds and precipitation region as defined by $C_{cld}$
threshold. These parameters for each cyclone event at band 13 were summarized in Table 2.
**3.3 Evaluation of normalized FG departures**
The bandwidth of FG departures are very high (Figure 4) and finding a symmetric bias in absolute
FG departure is not feasible. Hence, FG departures are normalized with $SD_{cloud}$ (eq. 4) at band 13.
The pdf of normalized FG departures were compared with Gaussian for all the deep convective
events (Figure 6). From Figure 6, it can be seen that, the normalized FG departure curves follow
symmetric distribution but its peak was too high with smaller errors. The main advantage of
symmetric error model is to assign large errors in cloudy conditions without causing difficulty in
all-sky assimilation.
**3.4 Quality Control (QC)**
Figure 7 (a), (b) and (c) shows the distribution between observed and simulated Tb using binned
scatter plots in 1.0 K by 1.0 K bin for hudhud, vardah and kyant cyclone respectively. Samples
found to be outside the range of 100-300 K and bins containing less than or equal to 1 sample were
removed from the analysis. The simulated warmer Tb (>240 K) was in good agreement with the





observations but samples containing low values of Tb  (<240 K) were either from first-guess or
from observations having large FG departures. Because of partially random distribution of deep
convective clouds, there is a large unceratinty in the prediction of exact location of convective
clouds in the model causes the large diagreement (Harnisch et al., 2016). Geer and Bauer, (2011)
proposed quality control (QC) method in operational all-sky microwave radiance assimilation to
eliminate the large FG departures due to cloud mis-location and instrumental errors, however, their
study has not considered the observations wherein normalized FG departures are greater than $\pm$
2.5 K.
For the present study at band 13, threshold limits cannot be decided using the normalized FG
departures. Hence, we performed QC by removing 2.5% samples from both sides of the tail of
normalized FG departures. Samples after QC are shown in Figure 7 (d), (e) and (f) and dashed line
represent the window of FG departure at $0, \pm 10$ and $\pm 30\,K$. The low Tb samples removed after
QC reduces the variability of FG departure and hence improves the symmetry. Mostly cloudy
samples were lie in the error range of $\pm 30\,K$. Results has shown the improvement in correlation
coefficient after QC. This method also eliminates the negative departures linked with deep
convective events. Figure 8 shows the convective clouds on 10[th] December 2016 at 03 UTC
wherein plots (a), (b) and (c) represent the observed and simulated Tb before and after QC
respectively. The cloud information remains preserved after the QC.

**3.5 Measure of goodness of fit**

Accurate simulation of deep convective events at 183 GHz are challenging due to difficulty in
modelling of scattering effects from frozen hydrometeors (Geer and Baordo, 2014; Guerbette et
al., 2016). This section measures the goodness of fit between observed and simulated radiances
using four widely used DDA shapes. It is common practice to use chi-squared or K-S test to
statistically measure the discrepancy between two distributions. Geer and Baordo,(2014) proposed
an 'h-statistics' (eq. 5) for smaller samples arranged into number of bins. The value of h could
reach infinite if no samples be present in the bin. This study assigns such bins to 0.1 value.
$$h = \frac{\left( \sum_{bins} \left| log \frac{\#\,simulated}{\#\,observation} \right| \right)}{total\,no\,of\,bins}$$    (5)
Here, the bin size is 2.5 K and # denote the numbers.



The Figure 9 (a), (b) and (c) shows the $log\frac{\#\ simulated}{\#\ observation}$ in y-axis for the case study events that lie
within the range of 100-280 K Tb in x-axis for all DDA shapes. In the 100-200 K bins, block
column and thin-plate have a positive log ratio which means low Tb occurs extensively due to
excessive scattering from clouds while six-bullet rosette and sector snowflake with negative log
ratio shows very less or no existence of Tb in this range because of insufficient scattering in heavy
cloud and precipitation regions. However, in 200-280 K bins, the log ratio closely lies near to zero
and overall good agreement is observed between the observed and simulated Tb. The analysis of
DDA shapes shows that thinplate has less peak from 0 to either positive or negative side in all the
cases. The h-statistics value for each shape is given in Table 3. Less the number, more will be the
similarity. Thin-plate have lowest h-value.
Figure 9 (d) shows the skewness of FG departure for all convective events for each DDA shapes.
Large negative or positive value indicate skew towards left or right from normal distribution curve.
Combining h-statistics and skewness, thin plate show optimum results among all DDA shapes over
Bay of Bengal at band 13. This result is in accordance with the study by Rysman et al., (2016)
which shows that thin plate perform best in simulation of all-sky radiances of Advanced
Microwave Sounding Unit (AMSU)-B sounding channels in Mediterranean region. Guerbette et
al., (2016) observed the best simulations of all-sky SAPHIR radiances with block column shape
over Indian Ocean at 183 $\pm$ 7 GHz.
**3.6 Sensitive to ERA-5 reanalysis datasets**
Simulated Tb using ERA-I datasets were compared with ERA-5 (31 km; 3 hr) simulation for all
cases. Figure 10 shows the spatial distribution of observed and simulated Tb using DDA thinplate
shape for convective clouds at (a, b, c) 10[th] October 2014-18 UTC (hudhud), (d, e, f) 8[th] December
2016-15 UTC (vardah) and (g, h, i) 22[nd] October 2016-18 UTC (kyant). It was observed that the
location of clouds and their intensity with ERA-5 datasets was much similar to observations and
clear mis-match in distribution of clouds was shown with ERA-I datasets.
The observed and simulated Tb is represented using box-plots and histogram in Figure 11 (a) and
(b). The total number of samples are 2445 and correlation coefficient has improved drastically
from 0.04 (ERA-I) to 0.52 (ERA-5). The ERA-I simulations have large variability of low Tb due





to excess scattering from clouds and also decreases the median value. Overall results state that
ERA-5 improves the displacement of cloud location, pattern and intensity.
**4    Summary and Conclusions**
The present study evaluates the simulation of all-sky microwave radiances of GPM/GMI using
Weather Research Forecast (WRF) and RTTOV-SCATT (v12.1) radiative transfer model. GMI
observations at water vapour sounding channel ($183 \pm 7$ GHz-V) has been considered and
spatially averaged over model resolution. This study has been conducted for three tropical cyclones
(Hudhud : $07^{th}$ -$14^{th}$ Oct. 2014; Vardah: $21^{st}$ -$29^{th}$ Oct. 2016 and Kyant: $06^{th}$ -$12^{th}$ Dec. 2016) at 3-
h interval and 15 horizontal resolution over Bay of Bengal (BOB) region. In the present study, four
recognized DDA shapes (sector snowflake, block column, thin plate and six-bullet rosette) were
considered for simulation of brightness temperature (Tb). Results show that simulations using mie-
spheres produces bias in cloudy region due to inadequate scattering at $183 \pm 7$ GHz while all DDA
shapes have significant scattering at higher frequency (Geer and Baordo, 2014) .
We evaluate the cloud effect on FG departures from DDA sector snowflake simulations using
symmetric error model. The probability distribution function of normalized FG departures are
found to be symmetric. The results show that cloudy samples can offer potential to be assimilated
in all-sky radiance assimilation experiments.
The present study also conducted the statistical measures to evaluate the performance of DDA
shapes for radiance simulation. The h-statistics is performed to measure the consistency between
observed and simulated distributions. We also used skewness, the most suitable parameter in large
errors situations (Wilks, 2006). This study observed that thinplate simulates all-sky microwave
radiances consistently with observations over BOB region. Our finding resonate with Rysman et
al., (2016) over mediterranean region at $183 \pm 7$ GHz. In our simulations, we consider only DDA
shapes for snow, however in reality there are also high density particles such as hail, aggregate
and graupel hydrometeors which produces very low brightness temperature (Figure 1). Further
efforts should be made to include varieties of frozen hydrometeros in RTTOV model. Another
improvement can be done in RTTOV-SCATT model to allow multiple DDA shapes of frozen
hydrometerors at a time.



In the present study, simulation of all-sky GMI radiances is carried out with ERA-1 and ERA-5
reanalysis datasets using thinplate shape. Results show improvement in cloud location and
intensity near the core when using ERA-5 compared to ERA-I datasets. This can be attributed to
the higher spatial and temporal resolution of ERA-5 datasets which when used in WRF model
improved the forecast of cloud and precipitation. The initial results using ERA-5 datasets are
encouraging and will be a part of ongiong work on radiance simulations. Future work will be
focussed on all-sky GMI radiance assimilation in WRF model at higher frequencies for short range
forecast over indian region.
**Acknowledgement**
This work was supported by the Indian Institute of Technology (I.I.T) Bombay, Powai under the
Project 15IRCCSG016. We are thankful to the NCAR for the WRF model. The first author would
like to thank Dr. James Hocking, Met Office, UK for initial help related to RTTOV-SCATT
Model. We would like to thank the Goddard Distributed Active Archive Center (GES DISC
DAAC) for providing the GMI radiances and are free available at https://mirador.gsfc.nasa.gov/.
The ERA-I and ERA-5 datasets are obtained from http://rda.ucar.edu/datasets/ds627.0 and
https://rda.ucar.edu/datasets/ds630.0/ respectively.

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



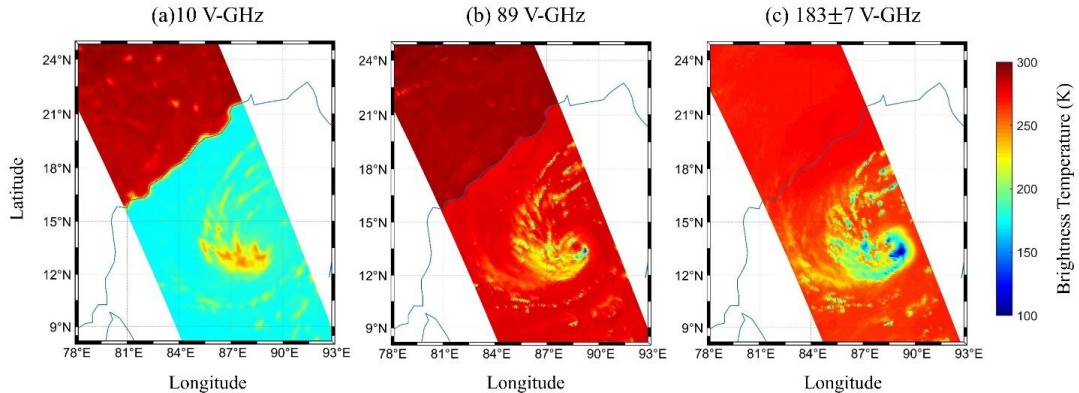


**Figure 1: Brightness Temperature from GPM GMI for (a) 10 V-GHz, (b) 89 V-GHz and (c) 183 ± 7**

**GHz for hudhud cyclone event on 9th October 2018 at 06 UTC. The frozen hydrometeors information**

**are more enhanced at 183 ± 7 GHz frequency.**


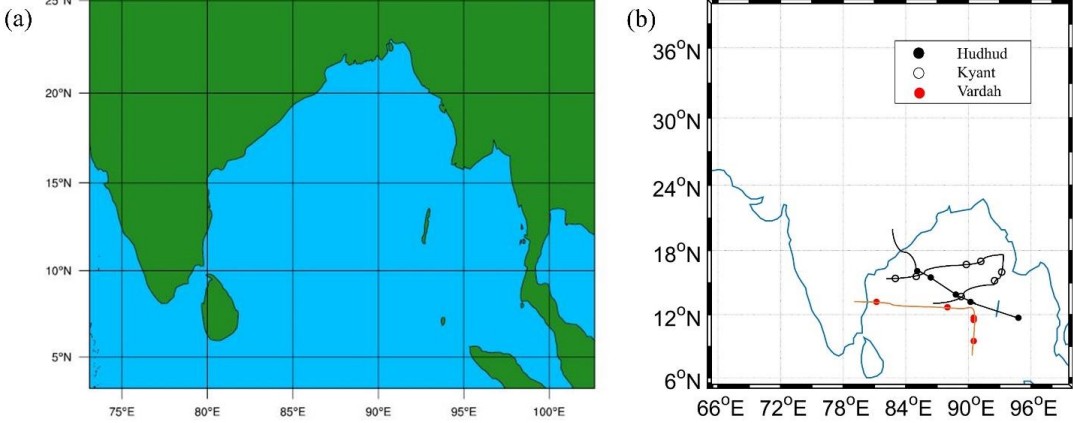


**Figure 2: (a) Single WRF domain used for the simulation of three Tropical cyclones (Hudhud,**

**Vardah and Kyant) over Bay of Bengal and (b) shows the track of cyclones and dot point represent**

**the availability of GMI observations near the eye of cyclone**








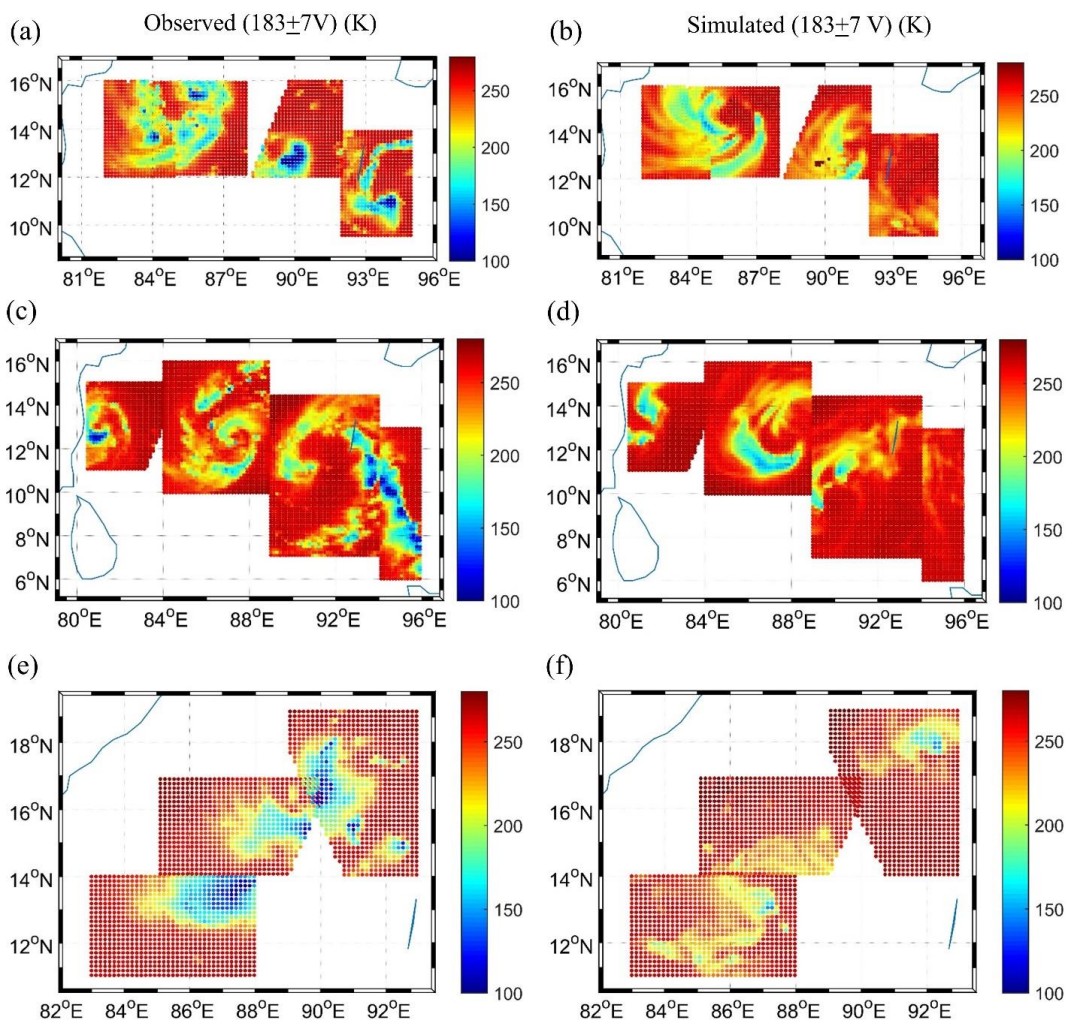


**Figure 3: Spatial distribution of (a, c and e) GMI observed brightness temperature (Tb) and (b, d and f) simulated Tb with default DDA sector shape at band 13 for hudhud, vardah and kyant cyclone respectively.**




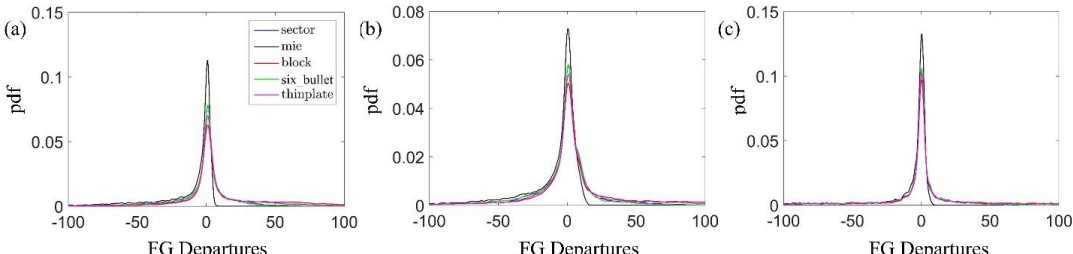

**Figure 4: Probability distribution function (PDF) of observed-background (FG departures) with mie-spheres and DDA shapes for (a) hudhud, (b) vardah and (c) kyant cyclone.**

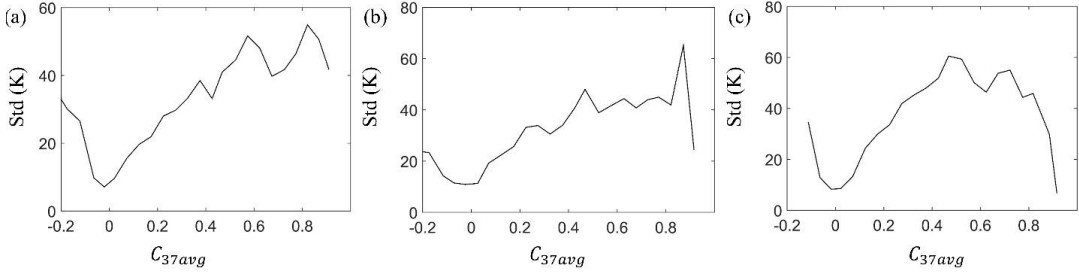

**Figure 5: Standard deviation ($SD_{cloud}$) curve with respect to average cloud amount at band 13 for (a) hudhud, (b) vardah and (c) kyant cyclone. The cloud amount bin is 0.05 at x-axis.**

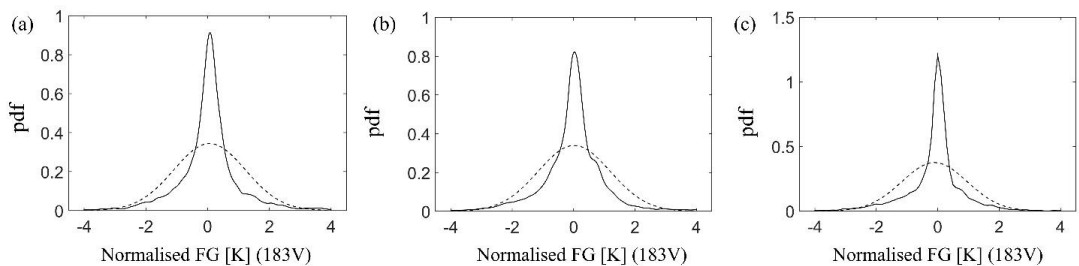

**Figure 6: Probability distribution function (PDF) of FG departure normalized by standard deviation as function of average cloud amount for (a) hudhud (b) vardah and (c) kyant cyclone at band 13. The dotted curve represent the Gaussian curve.**



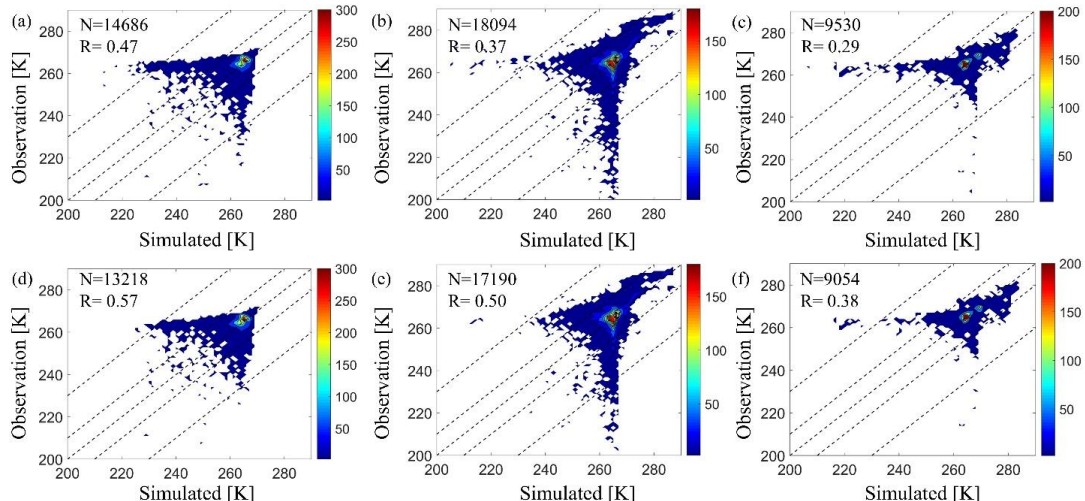

**Figure 7: Binned scatter plots of background and simulated brightness temperature at band 13 for (a, d) hudhud, (b, e) vardah and (c, f) kyant cyclone before and after Quality Control (QC). The samples are counted in 1.0 K by 1.0 K. The colorbar refers to the density in each bin. Dashed line represent the FG departure at 0, $\pm 10$ and $\pm 30$ K.**

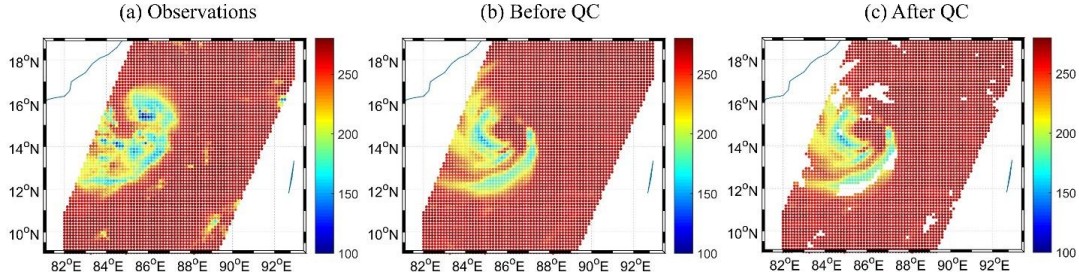

**Figure 8: shows the convective events at 10th October 2014:18 UTC represents (a) GMI observations, (b) Simulated Tb before QC, (c) after QC. The pixels of high FG departures due to mis-match of location were removed.**





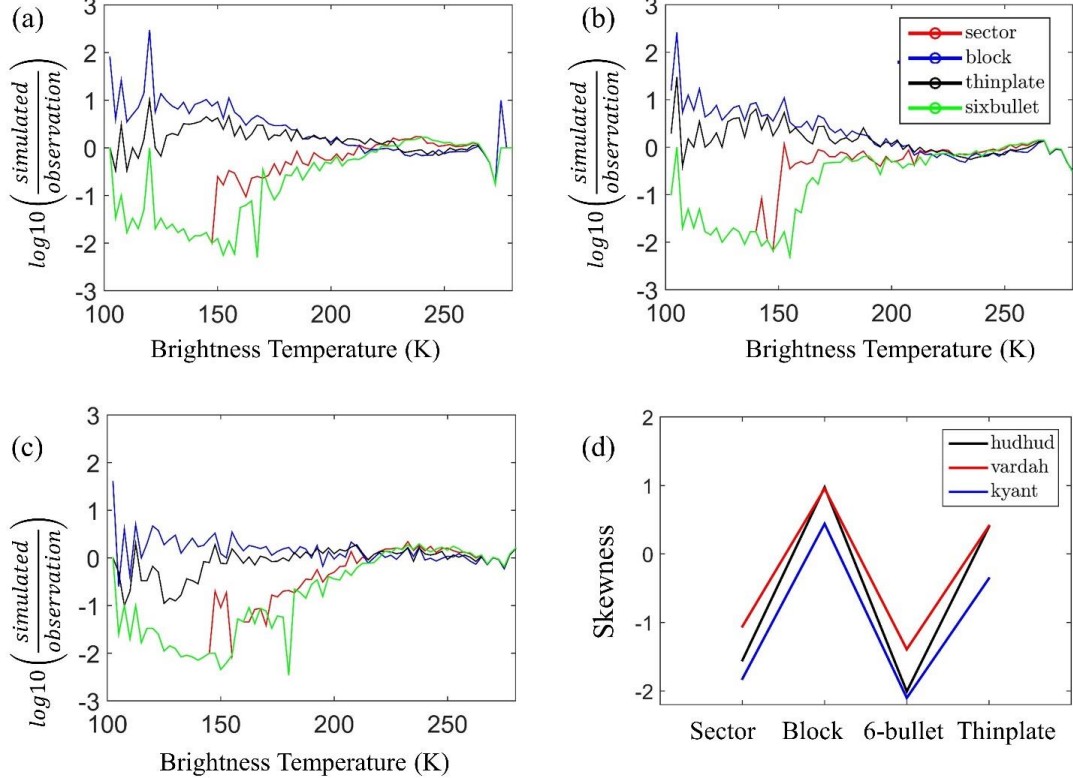

521

**Figure 9: Measure of goodness of fit. The log of the ratio of histograms (simulation divided by the observation) for four different DDA shapes over (a) hudhud, (b) vardah and (c) kyant cyclone. The bin size is 2.5 K. (d) represents the skewness of FG departures. Thinplate performs the best results over Bay of Bengal.**



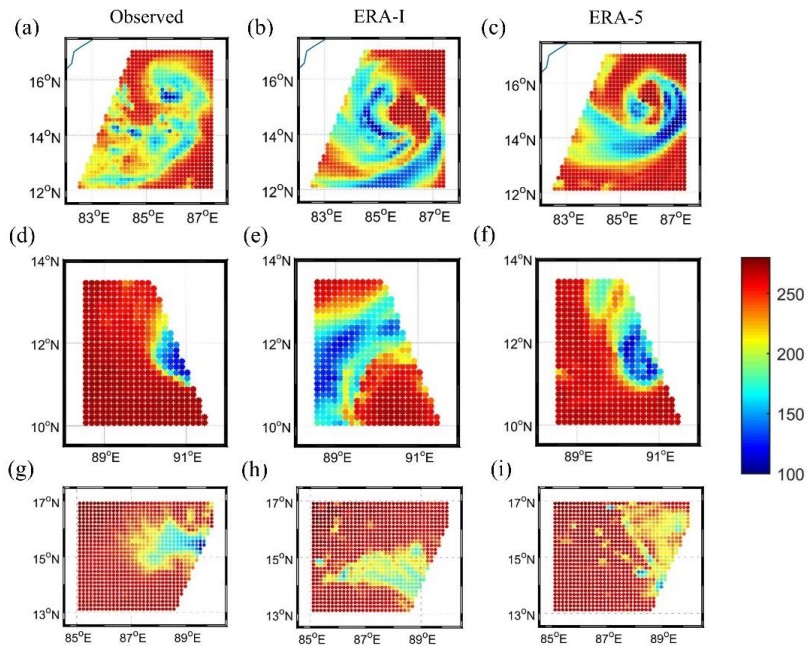

**Figure 10: Observed and Simulated Tb with ERA-I and ERA-5 reanalysis datasets for a day event (a, b, c) 10th October 2014, 18 UTC (hudhud cyclone); (d, e, f) 8th December 2016, 15 UTC (vardah cyclone) and (g, h, i) 22nd October 2016, 18 UTC (kyant cyclone).**

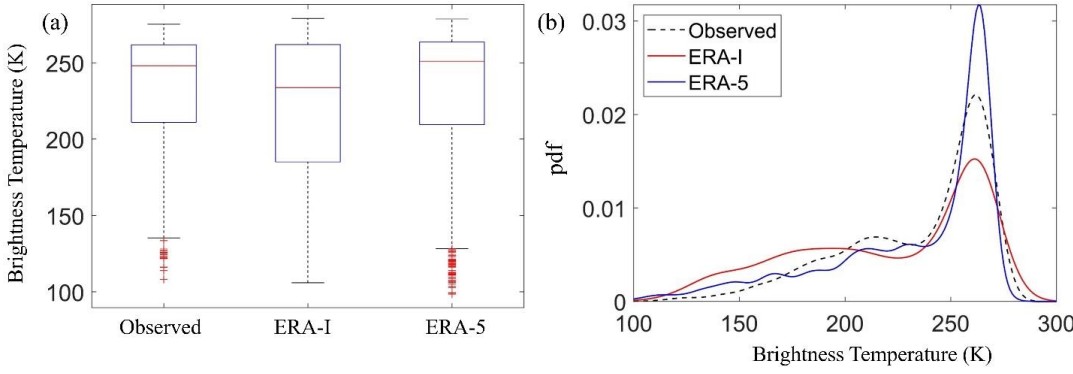

**Figure 11: (a) Boxplot of observed and Simulated Tb with ERA-I and ERA-5 reanalysis datasets. The 50 percentile of ERA-I simulations is larger than observed data due to excess scattering from the clouds. (b) Histogram of observed and simulated brightness temperature. ERA-5 simulations have similarity with observed data.**





**Table 1. GMI sensor characteristics (Hou et al., 2014)**

| Channels | Frequency (GHz) | Polarization | Resolution (Km) |
|----------|-----------------|--------------|-----------------|
| 1, 2 | 10.65 | V, H | 19.4 x 32.2 |
| 3,4 | 18.7 | V, H | 11.2 x 18.3 |
| 5 | 23.8 | V | 9.2 x 15.0 |
| 6,7 | 36.5 | V, H | 8.6 x 15.0 |
| 8,9 | 89.0 | V, H | 4.4 x 7.3 |
| 10,11 | 166 | V, H | 4.4 x 7.3 |
| 12 | 183$\pm$3 | V | 4.4 x 7.3 |
| 13 | 183$\pm$7 | V | 4.4 x 7.3 |



**Table 2. Standard deviation and threshold for identifying clear-sky and cloudy samples at 183 $\pm$ 7**
**V GHz for all tropical cyclones.**

| | Hudhud | Vardah | Kyant |
|---|--------|--------|-------|
| $S_{clr}$ | 9.5977 | 11.2895 | 8.9561 |
| $S_{cld}$ | 51.6524 | 47.9787 | 61.7815 |
| $C_{cld}$ | 0.5733 | 0.4677 | 0.5218 |
| $C_{clr}$ | 0.0209 | 0.0266 | 0.0213 |










**Table 3. shows the h-value and skewness corresponding to the all meteorological events for different**
**DDA shapes. Thinplate has the least h-value and low skewness in all events.**

| DDA shapes | Hudhud | | Vardah | | Kyant | |
|---|---|---|---|---|---|---|
| | h-value | skewness | h-value | skewness | h-value | skewness |
| Sector | 0.6297 | -1.5516 | 0.5477 | -1.0571 | 0.7565 | -1.8218 |
| Block | 0.4893 | 0.9733 | 0.4064 | 0.9533 | 0.2522 | 0.4399 |
| 6- bullet | 0.8171 | -2.0069 | 0.7028 | -1.3925 | 0.8838 | -2.1021 |
| Thinplate | 0.2743 | 0.3982 | 0.2528 | 0.4089 | 0.2397 | -0.3575 |
