# Peer review of "Evaluation of microwave radiances of GPM/GMI for the all-sky"

_Atmospheric Measurement Techniques, 2018_

## Referee Comment (RC1) · Anonymous Referee #1 · 4 Dec 2018

This paper describes a comparison between GMI microwave observations and radiative transfer simulations performed with the RTTOV-SCATT model and WRF forecasts as well as ERA-Interim and ERA-5 forecasts. Major work would be needed for this work to be considered for publication: the paper is poorly structured (e.g. WRF simulations are mentioned before explaining that WRF is used in the study ; the order of the paragraphs in section 3 does not make sense and it is not justified why ERA-Interim and ERA-5 datasets are used together with WRF without any obvious link between the two). Many statements are not justified by statistics but only by figures on which it is hard to see something. I would suggest to fully rewrite the paper with better selected results (either WRF only or ERA results) and figures supporting the findings.

[Figure]

Specific comments:

Introduction: Line 61: "Geer and Baordo, (2014) claimed that DDA sector snowflake is approximately fit for all frequencies at a global scale" => "Claimed" is not the appropriate word, the authors of this paper showed that the sector snowflake is a compromise over a large range of frequencies and over the globe with a large dataset.

Line 67: "FG departures" => FG has not been introduced before

Line 78: "Geer, (2013) used the same model at multiple frequencies of TMI and SSMIS channel." => The same model was not used, the parameters of it have been adapted to each sensor and each frequency.

Line 79: "In microwave spectrum, the symmetric error model is known to perform well for low frequencies" => The error model was used as well for high frequency, please read the papers around MHS all sky assimilation.

Line 82: "At higher microwave frequencies, the backscatter/brightness temperature registered by the sensor is mainly due to scattering from frozen hydrometeors, assuming a spherical shape" => Please do not mix backscatter and brightness temperature, they refer to two different things. The physical phenomena at play at high frequencies have nothing to do with "assuming a spherical shape", there is a confusion here between the physical phenomenon and the modeling of it. This sentence need to be rewritten.

Line 84: "Long-term monitoring of FG departure was found useful for identifying the instrumental error from ground based microwave observation (De Angelis et al., 2017)." => This sentence does not have much to do with the rest of the text. Please remove it.

Line 89: "In addition, we also include the analysis of ERA-5 reanalysis datasets (Malardel et 90 al., 2015) to extend the sensitivity to cloud physical processes at a higher resolution." => The authors mention ERA-5 but do not even explain before with which model they worked over India.

Data and method:

Line 109: "insensitive at 183 GHz." => This is a wrong statement. 183 GHz observations are sensitive to surface emissivity specification. This is why dynamical emissivity retrieval techniques have been developed for data assimilation.

Line 130: "The surface emissivity over oceans are calucated by the surface parameters" => calculated with

Paragraph 2.2 : the WRF model is mentioned here but is presented later, please restructure this section to present the information in a proper order

Results and Discussion:

Line 159 to 165: These details of the radiative transfer simulations should be given in section 2.2 Line 172: "upto Âã70-80 K" => down to 70-80K

Line 173: "Underestimation was observed using the mie-sphere, sector snowflake and six-bullet rosette shapes. Though the overall pattern and location of convective clouds near the eye of cyclone matched closely with the observations" => These statements are not supported by any statistics at this stage of the text

Line 182: "A negative departure occurs when the RTToV model is unable to produce realistic representations owing to cloud and precipitation." => This statement is wrong, a negative departure occur at 183 GHz when the simulated brightness temperatures are too warm with respect to the observation.

Line 183 to 187: "Within DDA shapes, the pdf curve is found to follow a symmetric distribution. ..." => The figure shown does not support any of this statement.

Line 188: "observation errros " => errors

Line 202: "As the quantities of . . . and ... are affected with sampling error (Geer and Bauer, 2011), their average is considered as the average cloud amount ..." => This sentence does not make sense.

Line 208: "The sudden peak at ... 0.8 " => These peaks are likely due to a smaller number of samples in these categories. The number of samples used is an important information to provide when trying to understand a result.

Line 217: "The bandwidth of FG departures are very high (Figure 4) and finding a symmetric bias in absolute FG departure is not feasible. " => This sentence does not make sense

Line 221: "From Figure 6, it can be seen that, the normalized FG departure curves follow symmetric distribution but its peak was too high with smaller errors. " => From the results of the normalization, it seems the distributions obtained are far from Gaussian. There must be a problem with the error model itself but unfortunately it is not shown in the paper.

3.4 Quality Control (QC) => Again the comments are not supported by statistics and it is hard to see the effect of the QC selected. There are also mistakes in the text, for instance normalized FG departures do not have a physical units like Kelvin (line 236). 3.5 Measure of goodness of fit => It does not make sense to have these results presented here after the error models. 3.6 Sensitive to ERA-5 reanalysis datasets => The results presented in this section are poorly introduced and have little connection with the results of the rest of the paper
* * *

---

## Referee Comment (RC2) · Anonymous Referee #2 · 18 Jan 2019

This study faithfully follows ECMWF's methodology (Geer et al , 2009, 2011,2013, 2014) on microwave radiance simulation affected by precipitation, specifically in frozen phase. The specific contribution to this research subject is to investigate one high frequency channel from GMI in one meteorological condition i.e. tropical storms in Bay of Bangal.

The following are specific comments to this very specific study:

(1) The model inputs to RTTOV are from WRF simulation of storms. It is not clear how WRF forecasts are conducted, e.g. are they straight many hours of simulation initialized by ERA from certain moment during the storm lifetime? Judging from the storm figures,

the model simulated precipitation spatial distributions are not good comparing to the observed, that in turn would have negative impact to sampling of Tb departures.

(2) The RTTOV simulation uses Field's PSD and particle density( Field 2007). However, the WRF simulation here uses one-moment 6-class microphysics (Hong and Lim 2006) that has different assumptions on PSD and density. This inconsistency can introduce model errors to the Tb departure statistics. As previous studies have shown, the simulated Tb in presence of frozen particles is sensitive to PSD and density assumption as much as to the non-spherical shape assumption.

(3) the WRF serves dynamic downscaling of ERA and introduces microphysics hydrometeors, However at the model resolution chosen for this study, microphysics is not as relevant and effective as the cumulus convective parameterization scheme. As a specific regional study a finer model resolution should be more appropriate to resolve what microphysics is designed to resolve, and avoid averaging Tb observations.

(4) The three storms presented are in the same region, same season, with similar meteorological characteristics. It would be a stretch to state that the study applies to "all meteorological conditions". Therefore the presentation of statistics should combine the samples from three storms, which can increase the sample size for each of the statistical parameters evaluated, and improve readable quality of the graphics presentation.

(5) With all respect to model simulations, I wish something could be done independent of ERA or WRF, i.e. find locally observed inputs to radiative transfer model to obtain Tb departures that do not contain model background errors. It is difficult to get that in general global study. It would be a much more meaningful contribution if a regional study like this one makes effort in that direction.

---

## Author Comment (AC1) · 13 Feb 2019

We thank reviewer 1 for their valuable suggestions to improve the quality of the manuscript. The critical evaluation is very helpful in improving our manuscript. Please find below the reviewer comments, followed by the author's response.

**General comment:**

This paper describes a comparison between GMI microwave observations and radiative transfer simulations performed with the RTTOV-SCATT model and WRF forecasts as well as ERA-Interim and ERA-5 forecasts. Major work would be needed for this work to be considered for publication: the paper is poorly structured (e.g. WRF simulations are mentioned before explaining that WRF is used in the study; the order of the paragraphs in section 3 does not make sense and it is not justified why ERA-Interim and ERA-5 datasets are used together with WRF without any obvious link between the two). Many statements are not justified by statistics but only by figures on which it is hard to see something. I would suggest to fully rewrite the paper with better selected results (either WRF only or ERA results) and figures supporting the findings.

**Response:**

Thank you for sharing the insightful comments and suggestions. In accordance with the suggestion, we have thoroughly rewritten the manuscript. We performed additional quantitative analysis as well as restructured the sections. Please find attached the specific point by point responses (R1) to the general comments (GC1)

**GC1:** "WRF simulations are mentioned before explaining that WRF is used in the study"

**R1:** As per reviewer suggestions, we have rearranged the order of sections 2.2 and 2.3. In the revised manuscript, section 2.2 explains the WRF NWP model (lines 128-156) and section 2.3 is RTTOV-SCATT v12.1 model (lines 158-193). For clarity, we incorporate the brief summary of WRF simulations in section 2.2 from lines 149 to 156.

**GC2:** "the order of the paragraphs in section 3 does not make sense"

**R2:** In consideration with the reviewer suggestion, the authors have rearranged the order of paragraphs.

In section 3, section 3.1-3.4 presents initial results in which WRF model is forced by ERA-Interim (ERA-I) reanalysis datasets and RTTOV model is configured with DDA sector snowflake shape for frozen hydrometeor. In section 3.1, we include Taylor diagram analysis to statistical justify the model performance from lines 201 to 222. In section 3.2, we add on the percentage of data points in addition to previous Figure 5 in order to get clarity of errors at high cloud amount (Lines 240-255). We replotted the Figure 6 with logarithmic y-axis and revised the discussion with additional analysis in section 3.3 (Lines 257-271). In section 3.4, the method for quality control (QC) is modified in revised manuscript. We changed the threshold value for QC and the new threshold was taken from Geer et al., (2014) for microwave humidity sounders at $183 \pm 7$ GHz. We included the root mean square parameter along with correlation coefficient for statistical comparison before

and after QC from lines 280 to 295. In section 3.5, we have re-run the RTTOV model with other DDA shapes for frozen hydrometeors without changing the WRF model configuration. Then, we performed the quantitative analysis using histogram, h-test and skewness to find the best shape (thinplate DDA shape) over Bay of Bengal region (lines 297-333). In section 3.6, we explain the reconfigured WRF model with new ERA-5 datasets. We have revised section 3.6 with more clarity to include Taylor diagram and boxplot diagram for statistical justification from lines 335 to 356.

**GC3:** "it is not justified why ERA-Interim and ERA-5 datasets are used together with WRF without any obvious link between the two"

**R3:** We utilize both ERA-I and ERA-5 datasets individually as forcing data and not together to compare the forcing input impact to radiance simulation. To add clarity, we add a new paragraph to introduce ERA-5 datasets in introduction section from line 89 to 97. WRF model is initially forced by ERA-I datasets and results are presented from section 3.1 to section 3.5. In section 3.6, we replace the forcing input with ERA-5 datasets and re-run the WRF model and the results were compared in radiance space as given in lines 335 to 356.

**Comment 1:** Introduction: Line 61: "Geer and Baordo, (2014) claimed that DDA sector snowflake is approximately fit for all frequencies at a global scale" => "Claimed" is not the appropriate word, the authors of this paper showed that the sector snowflake is a compromise over a large range of frequencies and over the globe with a large dataset.

**Response 1:** We replace 'claimed' word with 'suggested' in revised manuscript. The revised line 63 to line 64 states that : *"Geer and Baordo, (2014) suggested that DDA sector snowflake is approximately fit for all frequencies at a global scale but it doesn't perform well at the regional level."*

**Comment 2:** Line 67: "FG departures" => FG has not been introduced before

**Response 2:** In accordance with the comment, the explanation for FG departures have been included in lines 69 to 70 as: *"observed minus first guess or FG departures."*

**Comment 3:** "Geer, (2013) used the same model at multiple frequencies of TMI and SSMIS channel." => The same model was not used, the parameters of it have been adapted to each sensor and each frequency.

**Response 3:** In accordance with reviewer suggestions, the revised statement from line 80 to 81 now reads as *" The same parameters of error model are adapted at multiple frequencies of TMI, SSMIS (Geer, 2013) and MHS channels (Geer et al., 2014)."*

**Comment 4:** Line 79: "In microwave spectrum, the symmetric error model is known to perform well for low frequencies" => The error model was used as well for high frequency, please read the papers around MHS all sky assimilation.

**Response 4:** As per the comment, the lines 81 to 84 of the revised statement read as:

*"In microwave spectrum, FG departures are small and normalized PDF is close to gaussian for low frequencies (<50 GHz) as low frequencies are sensitive to column water vapur and rain-drops (Skofronick-Jackson and Wang, 2000) for which the particle shape and density are pre-defined."*

**Comment 5:** Line 82: "At higher microwave frequencies, the backscatter/brightness temperature registered by the sensor is mainly due to scattering from frozen hydrometeors, assuming a spherical shape" => Please do not mix backscatter and brightness temperature, they refer to two different things. The physical phenomena at play at high frequencies have nothing to do with "assuming a spherical shape", there is a confusion here between the physical phenomenon and the modeling of it. This sentence need to be rewritten.

**Response 5**: As per the reviewer suggestion, the revised statement is written from lines 84 to 86 as: *"At higher microwave frequencies, the brightness temperature (Tb) recorded at sensor is mainly due to scattering from frozen hydrometeors (Geer and Baordo, 2014) and FG departures are much higher (~±150 K) than lower frequencies."*

**Comment 6**: Line 84: "Long-term monitoring of FG departure was found useful for identifying the instrumental error from ground based microwave observation (De Angelis et al., 2017)." => This sentence does not have much to do with the rest of the text. Please remove it.

**Response 6**: We have removed this line from the revised manuscript.

**Comment 7**: Line 89: "In addition, we also include the analysis of ERA-5 reanalysis datasets (Malardel et al., 2015) to extend the sensitivity to cloud physical processes at a higher resolution." => The authors mention ERA-5 but do not even explain before with which model they worked over India.

**Response 7**: As per reviewer suggestions, we have added a new paragraph to introduce ERA-5 data and their importance in regional NWP model. They are added in introduction section from lines 89 to 96 as below:

*"Traditionally, regional NWP model such Weather Research Forecast (WRF) is forced by European Center for Medium Range Weather Forecast (ECMWF) ERA-Interim reanalysis datasets to provide initial and boundary conditions (Rysman et al., 2016; Singh et al., 2016). With the launch of high resolution ERA-5 reanalysis datasets over the period 2010-2016 by ECMWF*

*(Albergel et al., 2018a; Malardel et al., 2015), studies using Land Surafce Models (LSMs) have reported a significant improvement in the resulting simulated surface (soil moisture, snow depth etc) and atmospheric fluxes (Albergel et al., 2018b). Our study tries to examine the improvement in cloud representation in radiance space when WRF model is forced by ERA-5 datasets."*

**Comment 8:** Line 109: "insensitive at 183 GHz." => This is a wrong statement. 183 GHz observations are sensitive to surface emissivity specification. This is why dynamical emissivity retrieval techniques have been developed for data assimilation.

**Response 8:** we replace "insensitive" with "very low variation" in revised manuscript in line 118-120*:"The discrimination between land and ocean (Figure 1) is clear in 10 GHz, moderate in 89 GHz and very low variation at 183 GHz (Rysman et al., 2016)."*

**Comment 9:** Line 130: "The surface emissivity over oceans are calculated by the surface parameters" => calculated with

**Response 9**: In lines 168-170, we replace "calculated by" with "calculated with" in revised manuscript stated as*: "The surface emissivity over oceans are calculated with the surface parameters (eg. Temperature, surface wind) using the microwave surface emissivity model (FASTEM-version 6) (Kazumori and English, 2015)."*

**Comment 10:** Paragraph 2.2: the WRF model is mentioned here but is presented later, please restructure this section to present the information in a proper order.

**Response 10:** In accordance with reviewer suggestion, we rearrange the order of sections. Now, section 2.2 (lines 128-157) explains the WRF NWP model.

*In the revised manuscript, Lines 128 to 141 are stated as: "The WRF is specifically designed for regional forecast in operational and research NWP centers (Skamarock et al., 2008). The present study used the version 3.8 of WRF model for the forecasting of tropical cyclones over Indian region. We designed the experimental setup in a single domain from $3^oN\ to\ 26^oN$ and from $73^o\ E\ to\ 103^oE$ with 213x165 horizontal grids [Figure 2 (a)]. We selected the 15 km model resolution grid as compromise between the small number of samples and less effect of horizontally correlated errors (Liu and Rabier, 2002). This experiment is configured with 51 number of vertical layers and model top is at 125 hPa. The initial and boundary conditions are taken from ERA-Interim reanalysis datasets (product of ECMWF) with specification of 71 km spatial resolution at 6 h interval (Simmons et al., 2007). Geographical parameters including land use land cover (LULC), topography, soil type, lake and vegetation parameters are provided by the United States Geological Survey (USGS) global datasets at 30 sec resolution. Three tropical cyclones named*

"Hudhud" (October7-14, 2014), "Vardah" (December 6-12, 2016) and "Kyant" (October 21-27, 2016) over Bay of Bengal (BOB) regions are considered in this study. Their tracks are shown in Figure 2 (b)."

*In revised manuscript Lines 142 to 148 are stated as: "The physical parameterization schemes used are as suggested by (Routray et al., 2016) over BOB region are; WRF single moment 6-class microphysics scheme (Hong and Lim, 2006), Kain-Fritsch convection scheme (Kain, 2004), Yonsei scheme for planetary boundary layer (Hong et al., 2006), Dudhia shortwave radiation scheme (Dudhia, 1989), rapid radiative transfer model scheme for long-wave radiation (Mlawer et al., 1997) and Noah land surface model scheme (Tewari et al., 2004). This configuration is highly versatile for the prediction of short range forecast over the Indian region (Kumar et al., 2014; Singh et al., 2016)."*

*In the revised manuscript Lines 149 to 156 stated as: "The present study used the WRF model in control run mode (without assimilation). The experiment is designed for each cyclone as per the timing of GMI observations. For e.g. GMI observation for hudhud cyclone is at 0005 UTC 07th October 2014, the model is initialized at 0000 UTC 06th October 2014 with ERA-I datasets and then integrated up to 30 hours (06 UTC 7th October 2014) and the boundary conditions regularly updated at every 6h intervals from ERA-I datasets. The same procedure is repeated again and again for other GMI observations. In all the experiments, the analysis were generated within ±3 hour of the observations for every pass during the cyclonic event."*

Section 2.3 explains the RTTOV-SCATT v12.1 model as given below from 158 to 193 in revised manuscript.

*Lines 158 to 170 stated as: "The all-sky GMI radiances have been simulated using RTTOV-SCATT (version 12.1) of Radiative Transfer for the television infrared observation vertical sounder (RTToV) (Hocking et al., 2017; Saunders et al., 2017). The RTToV is initially developed by ECMWF which was then upgraded within the European Organization for the Exploitation of Metereological Satellites (EUMETSAT) NWP satellite application facility. This Model is suitable for rapid transformation of a huge number of NWP model outputs into the radiance space. The RTTOV-SCATT is a separate interface for the simulation of cloud and precipitation affected microwave radiances. As an input to RTTOV-SCATT model, the atmopheric profiles (i.e. temperature, water vapour, cloud liquid water, ice, snow and rain) were derived from WRF NWP model output. The delta-eddington approximation is used for solving the radiative transfer equations to simulate the scattering effects of clouds and precipitations (Joseph et al., 1976). The surface emissivity over oceans are calculated with the surface parameters (eg. Temperature, surface wind) using the microwave surface emissivity model (FASTEM-version 6) (Kazumori and English, 2015)."*

*Lines 171 to 193 stated as: "Generally, the bulk optical properties of clouds are hardly provided by NWP model which is mostly diagnostic in nature not prognostic (Doherty et al., 2007). It would be preferable that the RT model assumptions are consistent with NWP model but practically it is*

*not possible. Di Michele et al., (2012) suggested for independent assumptions to both NWP and RT models because of following reasons. (1) the assumptions related to parametrization (clouds and radiation) are not consistent within NWP model, (2) uncertainty in representing the spatial and temproral variation in PSDs and scattering information accurately in NWP model. However, independent assumptions can introduce model errors which is very small (~2 K) as compared to large systematic errors (~20-40 K) in convective environment (Geer and Bauer, 2011). Hence we ignore the model errors in all-sky radiance assimilation framework. We assumed the density of hydrometeors for rain and cloud liquid water=1000 kg/m3; ice=917 kg/m3; snow= 50 kg/m3) and Marshall-palmer distribution for rain, modified-gamma distribution for cloud liquid water and cloud ice. For frozen hydrometeors, it would be preferable to vary PSD and DDA shapes together for accurate scattering properties but this will be complicated for RTTOV model (Geer and Baordo, 2014). In the present study, we decided to fix PSD and density and keep varying the DDA shapes of frozen hydrometeor shape for Tb simulations. In literature, (Doherty et al., 2007; Kulie et al., 2010), Field et al., (2007) is well known PSD for frozen hydrometeors over tropical and mid-latitude regions. Hence we selected the same PSD over Indian region.*

*The all-sky Tb computed represent the weighted summation of the clear and cloudy independent columns (eq. 1). The weighing criteria is decided by the cloud fraction (Geer et al., 2009) which is based upon the variation in cloud and precipitation at subgrid scale.*

$$Tb^{all-sky} = C_f * Tb^{cloudy} + \left(1 - C_f\right) * Tb^{clear-sky} \qquad (1)$$

*Here, $C_f$ represents the vertical profile of cloud fraction.*"

**Comment 11:** Line 159 to 165: These details of the radiative transfer simulations should be given in section 2.2. Line 172: "upto ¢a70-80 K" => down to 70-80K.

**Response 11:** As per the reviewer suggestion, the details of radiative transfer simulations are shifted to section 3.3 (lines 180 to 188) in the revised manuscript. This has been addressed as part of comment 10. As suggested in line 208, we replace 'upto' word to 'down to' in revised manuscript.

**Comment 12:** Line 173: "Underestimation was observed using the mie-sphere, sector snowflake and six-bullet rosette shapes. Though the overall pattern and location of convective clouds near the eye of cyclone matched closely with the observations" => These statements are not supported by any statistics at this stage of the text.

**Response 12:** In order to justify our results, we have performed additional analysis using Taylor diagram and revised the whole paragraph as per reviewer suggestions. Lines 201 to 222 of the revised manuscript now reads:

*Lines 201 to 213 as: "The Figure 3 shows comparison between the all-sky simulated radiances at band 13 with respect to the observed GMI radiances for three cyclonic events over the BOB region. The microwave observations were averaged to 15 km horizontal resolution to match closely with the effective resolution of NWP model. Overall, simulation results matches well with observed radiances, however simulated Tb is warmer than observed (indicate underestimation) in some cloud or precipitation regions. The dominancy of scattering properties from frozen hydrometeors at band 13 in deep convective zones result in low temperatures of observed radiances inside the core of cyclone (down to 70-80 K). This may be attributed to deficiency of frozen hydrometeors at sub-grid scale in Kain-Fritsch convection scheme (Rysman et al., 2016). A study by Wu et al., (2015) found that frozen hydrometeors are underestimated in WRF simulations by all convective parametrization schemes over central and eastern pacific region. Rysman et al., (2016) estimate the underestimation in WRF simulations by a factor of 5 using airborne radar in Mediterranean region."*

*Lines 214 to 222 stated as: "In this study, the precise location and intensity of cloud and precipitation is not being captured. One of the possible reasons might be model tendency to predict the cloud earlier than observation in control run experiment (Yesubabu et al., 2016) which causes large FG departures. These departures are called as representative or observation errors in operational all-sky assimilation. Taylor diagram in Figure 4 provides the brief summary of model performance for all the events. The correlation coefficient ranges from 0.3 to 0.45 and root mean square error differences ranges from 23-26 K. The standard deviations (represents variability) indicate that simulation variability is approximately less than 10 K than observation variability. The detailed statistics is given in Table 2. Overall, the performance of model is quite consistent over Bay of Bengal region."*

[Figure]

**Figure 1: Spatial distribution of (a, c and e) GMI observed brightness temperature (Tb) and (b, d and f) simulated Tb with default DDA sector shape at band 13 for hudhud, vardah and kyant cyclone respectively.**

[Figure]

**Figure 2 Taylor diagram for observed and simulated Tb for hudhud, vardah and kyant cyclonic event. Results shows the robust nature of model for Tb simulation at higher frequency.**

**Table 1 Taylor results for simulation of GMI radiances for Hudhud, Vardah and Kyant cyclone**

|  |  | Observed Tb | Simulated Tb |
|---|---|---|---|
| Correlation (r) | Hudhud | 1 | 0.4518 |
|  | Vardah | 1 | 0.3684 |
|  | Kyant | 1 | 0.3209 |
| RMSD | Hudhud | 0 | 23.5366 |
|  | Vardah | 0 | 25.6074 |
|  | Kyant | 0 | 26.4866 |
| Standard Deviations | Hudhud | 25.1173 | 18.5506 |
|  | Vardah | 25.5403 | 19.00 |
|  | Kyant | 27.0963 | 15.2469 |

**Comment 13**: Line 182: "A negative departure occurs when the RTToV model is unable to produce realistic representations owing to cloud and precipitation." => This statement is wrong, a negative departure occur at 183 GHz when the simulated brightness temperatures are too warm with respect to the observation.

**Response 13**: We revised the statement in accordance with suggestion from line 302 to line 303 as follows: *"A negative departure occurs when the RTTOV simulation is warmer than observation."*

**Comment 14**: Line 183 to 187: "Within DDA shapes, the pdf curve is found to follow a symmetric distribution. ..." => The figure shown does not support any of this statement.

**Response 14**: As per reviewer comment, we have re-plotted the figure 9 with logarithmic y-axis in revised manuscript. Geer, (2013) stated "*If the disagreement between model and observations is due to random forecast errors, the histogram should be symmetric*". In our study, the FG departures of DDA based simulations spread equally in positive as well as negative direction due to random forecast errors. Hence, departures are symmetrical in nature. Our results are in accordance with Geer, (2013). The detailed explanation of figure 9 is given from line from 300 to 309 as below:

*"Figure 9 shows histogram of FG departures in mie-sphere and DDA shapes (sector snowflake, block column, six-bullet rosette and thinplate) for all the events. A negative departure occurs when the RTTOV simulation is warmer than observation. If the model predict precipitation and observation does not, the FG departures within DDA shapes spreads equally in both directions due to random forecast errors from first-guess and observations and tails of negative and positive departures are very similar to each other. In case of mie-spheres (black curve), the shift towards large negative departures indicates the presence of bias in cloudy region. This is because of insufficient scattering by mie-spheres at band 13 (Geer, 2013; Geer and Baordo, 2014). Results show that DDA simulations provide a better realistic scattering in all-sky conditions and more symmetrical FG departures."*

[Figure]

**Figure 3. Histogram of observed-background (FG departures) with mie-spheres and DDA (sector snowflake, block column, six-bullet rosette and thinplate) shapes for (a) hudhud, (b) vardah and (c) kyant cyclone. The y axis is logarithmic.**

**Comment 15:** Line 188: "observation errros " => errors

**Response 15:** In line 223, we corrected the 'errros' to 'errors' in revised manuscript.

**Comment 16:** Line 202: "As the quantities of . . . and ... are affected with sampling error (Geer and Bauer, 2011), their average is considered as the average cloud amount ..." => This sentence does not make sense.

**Response 16:** As per reviewer suggestion, line 235 to 236 shows the revised statement as given below. *"This study used the average of observed and simulated cloud amount ($C_{37avg}$) to avoid sampling error as suggested by Geer and Bauer, (2011)."*

**Comment 17:** Line 208: "The sudden peak at . . . 0.8 " => These peaks are likely due to a smaller number of samples in these categories. The number of samples used is an important information to provide when trying to understand a result.

**Response 17**: In accordance with reviewer suggestions, we added the percentage of observations with respect to cloud amount in Figure 5. Then revised statement is given below from lines 253 to 255 as:

*"The sudden peak above threshold point ($C_{cld}$) in hudhud and vardah cyclone due to lower percentage of data points and their contribution is insignificant."*

[Figure]

**Figure 4. Standard deviation ($SD_{cloud}$) curve with respect to average cloud amount at band 13 for (a) hudhud, (b) vardah and (c) kyant cyclone. The cloud amount bin is 0.05 at x-axis.**

**Comment 18:** Line 217: "The bandwidth of FG departures are very high (Figure 4) and finding a symmetric bias in absolute FG departure is not feasible. " => This sentence does not make sense

**Response 18:** As reviewer suggested, we have removed the line 217 from revised manuscript.

**Comment 19:** Line 221: "From Figure 6, it can be seen that, the normalized FG departure curves follow symmetric distribution but its peak was too high with smaller errors." => From the results of the normalization, it seems the distributions obtained are far from Gaussian. There must be a problem with the error model itself but unfortunately it is not shown in the paper.

**Response 19:** As per reviewer suggestions, we have replotted the figure 6 with logarithmic y-axis and in addition, we include the normalized FG curve with constant standard deviation in Figure 6 in order to get more clarity of performance of symmetric error model. However, normalized PDF curve does not exactly match with Gaussian but in accordance with Geer, (2013). The detailed explanations are given in lines from 257 to 271.

*"This section evaluates the Gaussianity of the PDF of FG departures normalized by its standard deviation. The standard deviation was calculated in two ways: (1) consider all samples without any cloud parameter ($SD_{const}$) and (2) from the binned standard deviation in association with cloud amount ($SD_{cloud}$) from figure 5. The PDF of normalized FG departures were compared with Gaussian for all the deep convective events in Figure 6. It can be seen that, the normalized FG departures by $SD_{const}$ (blue color curve) is peaked and have wider tails because $SD_{const}$ is large for small departures and vice-versa. However, normalized PDF by symmetric error model (red color curve) is quite closer to Gaussian as compare to $SD_{const}$. Although, PDF is not perfectly Gaussian but error model reduces the skewness and the curve approaches to Gaussian. The extra peak in PDF suggests that errors are over cautious, as a lot of clear-sky observations have been assigned to large FG departures. In addition, clear-sky humidity errors are not considered in observation error model. The results are consistent with humidity sounding channels of SSMIS and MHS sensors (Geer, 2013; Geer et al., 2014). Similarly, humidity channel of infrared sensors also face same issue (not Gaussian) and suggested to include humidity and ozone parameter in error model in future work (Okamoto et al., 2014)."*

[Figure]

**Figure 5. Probability distribution function (PDF) of FG departure normalized by standard deviation as function of average cloud amount for (a) hudhud (b) vardah and (c) kyant cyclone at band 13. The dotted curve represent the Gaussian curve. The y axis is logarithmic.**

**Comment 20:** 3.4 Quality Control (QC) => Again the comments are not supported by statistics and it is hard to see the effect of the QC selected. There are also mistakes in the text, for instance normalized FG departures do not have a physical units like Kelvin (line 236).

**Response 20:** As per reviewer suggestion, we have altered the section 3.4. We change the threshold value for QC and the new threshold was taken from Geer et al., (2014) for microwave humidity sounders at $183 \pm 7$ GHz. We include the root mean square parameter along with correlation coefficient for statistical comparison before and after QC. Lines 280-295 are stated as

*"Geer and Bauer, (2011) proposed basic quality control (QC) or first guess check in operational all-sky microwave radiance assimilation to eliminate the large FG departures. Their study has eliminated the observations wherein normalized FG departures are greater than ± 2.5. Similarly, Geer et al., (2014) used the same approach for MHS sensor at 183 GHz frequency.*

*For the present study at band 13, we used the same threshold limit (± 2.5) of normalized FG departure and ~4-5% data has been eliminated in all the events. Samples after QC are shown in Figure 7 (d), (e) and (f) and dashed line represent the window of FG departure at 0, ±10 and ±30 K. Mostly cloudy samples were found to lie in the error range of ±30 K. Results have shown the improvement in correlation coefficient (increases by ~0.10) and root mean square error (decreases by ~4-5 K) after QC. The observations removed either due to inaccurate simulations from RT and NWP model or due to larger FG departures than expected (Geer et al., 2014). This basic QC method also eliminates the negative departures linked with deep convective events. Figure 8 shows the convective clouds on 10th October 2014 at 18 UTC wherein plots (a), (b) and (c) represent the observed and simulated Tb before and after QC respectively. The random cloud signatures in observed data are removed after QC, however cloud information remains preserved."*

The unit error in the text has removed in the revised manuscript.

**Comment 21:** 3.5 Measure of goodness of fit => It does not make sense to have these results presented here after the error models.

**Response 21:** The purpose of section 3. 5 is to identify the best DDA shape at higher frequency to simulate tropical cyclones over Bay of Bengal region. This study presents the initial results (section 3.1-3.4) using DDA sector snowflake shape. In section 3.5 (lines 297 to 333), shape factor is analyzed for Tb simulation in RTTOV model using recognized DDA shapes (sector snowflake, thinplate, block column and six-bullet rosette) of frozen hydrometeors.

**Comment 22:** 3.6 Sensitive to ERA-5 reanalysis datasets => The results presented in this section are poorly introduced and have little connection with the results of the rest of the paper

**Response 22:** As reviewer suggested, for clarity, we have introduced ERA-5 data in introduction section from line 89 to 97. In section 3.6, we reconfigured the WRF model with ERA-5 datasets and run again the RTTOV model. Then, we perform comparative analysis of both the results. We include Taylor diagram and box plot for statistical analysis and revised the text of whole section in revised manuscript as given below from lines 335 to 356. We also changed the title of section 3.6 "Sensitive to ERA-5 reanalysis datasets" to "Impact of high resolution atmospheric profiles on radiance simulation" for more clarity.

*Lines 335 to 344 stated as: "This section investigates the indirect impact of higher resolution forcing data (ERA-5; 31 Km at 3 h interval) in WRF NWP model on radiance simulation. It would be expected that higher resolution atmospheric profiles might have low uncertainty in determining cloud system with low Tb. The WRF model was reconfigured with new ERA-5 datasets and RTTOV-SCATT model was run with DDA thinplate shape of frozen hydrometeor. Simulated radiances at three convective events (Figure 11: (a-c) 10th October 2014:18 UTC (hudhud), (d-f) 8th December 2016: 15 UTC (vardah) and (g-i) 22nd October 2016:18 UTC (kyant)) were compared with observed and previous ERA-I based simulation. It was observed that the location of clouds and their intensity with ERA-5 datasets was much similar to observations and clear mismatch in distribution of clouds was shown with ERA-I datasets. "*

*Lines 345 to 356 stated as: "Figure 12 compared the FG departures from ERA-I and ERA-5 simulations using boxplot diagram. In ERA-I simulations, the range of FG departures are high and median value is negative, while with ERA-5 simulations, the FG departures range is quite low and median value approaches to zero in all cases. The excess scattering from frozen hydrometeors would be the reason of large variability of low Tb in ERA-I simulations. However, the variation in kyant event is not much higher because of similar cloud pattern as shown in Figure 11, only the difference is in mis-match of location. Taylor diagram (Figure 13) summarized the correlation coefficient, rmsd and standard deviations for all cases. The number of samples are 1071, 448 and 927 for hudhud, vardah and kyant cyclone event respectively. The correlation coefficient has improved, majorly in vardah event from -0.02 to 0.78, rmsd (59 K to 26.18 K) and variability (48.23 K to 42.09 K) has been decreased with ERA-5 simulations. The complete statistics is given in Table 5. Overall results state that ERA-5 improves the displacement of cloud location, pattern and intensity."*

[Figure]

**Figure 6. Observed and Simulated Tb with ERA-I and ERA-5 reanalysis datasets for a day event (a, b, c) 10th October 2014, 18 UTC (hudhud cyclone); (d, e, f) 8th December 2016, 15 UTC (vardah cyclone) and (g, h, i) 22nd October 2016, 18 UTC (kyant cyclone).**

[Figure]

**Figure 7. Boxplot of FG departures from ERA-I and ERA-5 simulations over hudhud, vardah and kyant cyclone. Box boundaries are 25th-75th percentiles. The 50th percentile of FG departures from ERA-I simulations is larger than ERA-5 because of excess scattering from clouds within Thinplate shape.**

[Figure]

**Figure 8 Taylor diagram showing the statistical comparison of simulated radiance using ERA-I and ERA-5 datasets. Overall correlation and RMSD has been improved.**

**Table 2 Summary of Taylor plot results comparing the ERA-I and ERA-5 simulations**

|  |  | Observed Tb | Simulated Tb (ERA-I) | Simulated Tb (ERA-5) |
|---|---|---|---|---|
| Correlation | 10th October 2014, 18 UTC | 1 | -0.2091 | 0.3558 |
|  | 8th December 2016, 15 UTC | 1 | -0.0231 | 0.7831 |
|  | 22nd October 2016, 18 UTC | 1 | 0.1408 | 0.4104 |
| RMSD | 10th October 2014, 18 UTC | 0 | 58.7170 | 43.9708 |
|  | 8th December 2016, 15 UTC | 0 | 59.0069 | 26.1804 |
|  | 22nd October 2016, 18 UTC | 0 | 36.7586 | 28.8519 |
| Standard deviations | 10th October 2014, 18 UTC | 31.5460 | 43.3649 | 43.8487 |
|  | 8th December 2016, 15 UTC | 32.8893 | 48.2375 | 42.0991 |
|  | 22nd October 2016, 18 UTC | 28.9102 | 27.1338 | 23.5853 |

---

## Author Comment (AC2) · 13 Feb 2019

We would like to thank the reviewer 2 for their efforts. The critical comments and questions have been very helpful for improving the manuscript.

**General Comments**

This study faithfully follows ECMWF's methodology (Geer et al, 2009, 2011, 2013, 2014) on microwave radiance simulation affected by precipitation, specifically in frozen phase. The specific contribution to this research subject is to investigate one high frequency channel from GMI in one meteorological condition i.e. tropical storms in Bay of Bengal.

**Comment 1:**

The model inputs to RTTOV are from WRF simulation of storms. It is not clear how WRF forecasts are conducted, e.g. are they straight many hours of simulation initialized by ERA from certain moment during the storm lifetime?

Judging from the storm figures, the model simulated precipitation spatial distributions are not good comparing to the observed that in turn would have negative impact to sampling of Tb departures.

**Response 1:**

As per the reviewer suggestions, for clarity we have stated the brief summary of WRF simulation experiment in the revised manuscript from line 149 to 156 as below:

*"The present study used the WRF model in control run mode (without assimilation). The experiment is designed for each cyclone as per the timing of GMI observations. For e.g. GMI observation for hudhud cyclone is at 0005 UTC 07th October 2014, the model is initialized at 0000 UTC 06th October 2014 with ERA-I datasets and then integrated up to 30 hours (06 UTC 7th October 2014) and the boundary conditions regularly updated at every 6h intervals from ERA-I datasets. The same procedure is repeated again for other GMI observations. In all the experiments, the analysis were generated within ±3 hour of the observations for every pass during the cyclonic event."*

Studies on Tb simulations (Fabry and Sun, 2010; Roberts and Lean, 2008; Rysman et al., 2016) show that it was very difficult to match the exact position and intensity of cloud and precipitation features in NWP models because of lack of cloud and precipitation predictability at smaller scales. In addition, Yesubabu et al., (2016) found that control run experiment in WRF model have tendency to predict the cloud earlier than observation. Tb departures are contributed by many factors including unknown size, shape and intensity of hydrometeors in modelling of cloud features which are significant around 20-40 K in convective environment (Geer and Baordo, 2014; Geer and Bauer, 2011). For clarity, we have included lines 214-216 in revised manuscript as follow.

*"In this study, the precise location and intensity of cloud and precipitation is not being captured. One of the possible reasons might be model tendency to predict the cloud earlier than observation in control run experiment (Yesubabu et al., 2016) which causes large FG departures."*

**Comment 2:**

The RTTOV simulation uses Field's PSD and particle density (Field 2007). However, the WRF simulation here uses one-moment 6-class microphysics (Hong and Lim 2006) that has different assumptions on PSD and density. This inconsistency can introduce model errors to the Tb departure statistics. As previous studies have shown, the simulated Tb in presence of frozen particles is sensitive to PSD and density assumption as much as to the non-spherical shape assumption.

**Response 2:**

We agree with the reviewer that WRF simulation uses different assumptions for PSD and density and that this can introduce model errors into the Tb departure statistics. But, the model error is very small (~2 K) as compare to large observation errors (~20-40 K) in convective environment (Geer and Bauer, 2011). Hence we can ignore the model errors in all-sky radiance assimilation framework. For clarity, we have included line173 to 179 as given below:

*"Di Michele et al., (2012) suggested for independent assumptions to both NWP and RT models because of following reasons. (1) the assumptions related to parametrization (clouds and radiation) are not consistent within NWP model, (2) uncertainty in representing the spatial and temproral variation in PSDs and scattering information accurately in NWP model. However, independent assumptions can introduce model errors which is very small (~2 K) as compared to large systematic errors (~20-40 K) in convective environment (Geer and Bauer, 2011). Hence we ignore the model errors in all-sky radiance assimilation framework."*

In accordance with reviewer comment, due to model complexity, varying PSD and particle shape together is not an feasible solution as stated below in lines 183-188.

*"For frozen hydrometeors, it would be preferable to vary PSD and DDA shapes together for accurate scattering properties but this will be complicated for RTTOV model (Geer and Baordo, 2014). In the present study, we decided to fix PSD and density and keep varying the DDA shapes of frozen hydrometeor shape for Tb simulations. In literature, (Doherty et al., 2007; Kulie et al., 2010), Field et al., (2007) is well known PSD for frozen hydrometeors over tropical and mid-latitude regions. Hence we selected the same PSD over Indian region."*

*."*

**Comment 3:**

The WRF serves dynamic downscaling of ERA and introduces microphysics hydrometeors, However, at the model resolution chosen for this study, microphysics is not as relevant and effective as the cumulus convective parameterization scheme. As a specific regional study a finer model resolution should be more appropriate to resolve what microphysics is designed to resolve, and avoid averaging Tb observations.

**Response 3:**

We thank the reviewer for the useful suggestions. If we select finer resolution (for e.g. 5 km), there will be high possibility of horizontally correlated errors and this may also increases the computational effort (Liu and Rabier, 2002). Moreover, past studies on all-sky radiance assimilation has been performed at 15 km model resolution for cyclonic event (Yang et al., 2016). Hence, we selected 15 km in our study.

For clarity, we include lines 131-133 in revised manuscript as below:

*"We selected the 15 km model resolution grid as compromise between the small number of samples and less effect of horizontally correlated errors (Liu and Rabier, 2002)."*

We agree with reviewer suggestions. Finer resolution (grid size < 10 km) can resolve the mixed phase processes that can produce large grapual and hail, particular in convective situations (Skamarock et al., 2008). But, RTTOV model version 12.1 does not consider grapual and hail and also their occurrence is very less over Indian region as compared to US and European regions (Baker et al., 2005). Hence, the present work does not consider finer resolution.

*Lines 382 to 386 stated as: "In our simulations, we consider only DDA shapes for snow, however in reality there are also high density particles such as hail, aggreagate and graupel hydrometeors which produces very low brightness temperature (Figure 1). Further efforts shall be directed towards including varieties of frozen hydrometeors in RTTOV model."*

**Comment 4:**

The three storms presented are in the same region, same season, with similar meteorological characteristics. It would be a stretch to state that the study applies to "all meteorological conditions". Therefore the presentation of statistics should combine the samples from three storms, which can increase the sample size for each of the statistical parameters evaluated, and improve readable quality of the graphics presentation.

**Response 4:**

We thank the reviewer for this suggestion. Though combining samples from three storms is a good approach but our hypothesis is to independently analyze the simulations of each cyclonic event and we intent to utilize the same for RTM configuration during assimilation of GMI radiances for better forecasting of each event. Further, previous studies on radiance assimilation have adopted similar techniques to evaluate the independent storm events (Madhulatha et al., 2017; Routray et al., 2016; Yang et al., 2016). Hence, for clarity we have removed the following lines from the revised manuscript" all meteorological conditions. "

**Comment 5:**

With all respect to model simulations, I wish something could be done independent of ERA or WRF, i.e. find locally observed inputs to radiative transfer model to obtain Tb departures that do not contain model background errors. It is difficult to get that in general global study. It would be a much more meaningful contribution if a regional study like this one makes effort in that direction.

**Response 5:**

Thank you for encouraging us with new suggestions. Existing studies used ERA-Interim (Zou et al., 2016), NCEP/NCAR reanalysis datasets (Mohanty et al., 2010) directly as input to RTTOV model for Tb simulations in clear-sky conditions. However, these datasets have lack of cloud ice and rain properties which cannot simulate the scattering properties accurately in all-sky conditions (He et al., 2016). Hence, we do not use this direct approach at this stage and consider it extremely useful for future studies in all-sky situations. For clarity, we have included 393-397 lines in revised manuscript are stated as:

*Moreover, independent studies with ERA datasets are not found useful for simulation especially for microwave humidity sounders because scattering properties of cloud and precipitation are not simulated accurately in all-sky conditions (He et al., 2016). This happens due to lack of accurate information about cloud ice and rain profiles in ERA-I datasets."*

Our future work is directed towards assimilating all-sky GMI radiances in WRF model to improve the initial conditions which reduces the model background errors and Tb departures.